# De novo human genome assemblies reveal spectrum of alternative haplotypes in diverse populations

Karen H.Y. Wong [1], Michal Levy-Sakin[1] & Pui-Yan Kwok [1,2,3]

The human reference genome is used extensively in modern biological research. However, a single consensus representation is inadequate to provide a universal reference structure because it is a haplotype among many in the human population. Using 10× Genomics (10×G) "Linked-Read" technology, we perform whole genome sequencing (WGS) and de novo assembly on 17 individuals across five populations. We identify 1842 breakpoint-resolved non-reference unique insertions (NUIs) that, in aggregate, add up to 2.1 Mb of so far undescribed genomic content. Among these, 64% are considered ancestral to humans since they are found in non-human primate genomes. Furthermore, 37% of the NUIs can be found in the human transcriptome and 14% likely arose from *Alu*-recombination-mediated deletion. Our results underline the need of a set of human reference genomes that includes a comprehensive list of alternative haplotypes to depict the complete spectrum of genetic diversity across populations.

---

[1] Cardiovascular Research Institute, University of California, San Francisco, San Francisco, 94158 CA, USA. [2] Institute for Human Genetics, University of California, San Francisco, San Francisco, 94143 CA, USA. [3] Department of Dermatology, University of California, San Francisco, San Francisco, 94115 CA, USA. Correspondence and requests for materials should be addressed to P.-Y.K. (email: pui.kwok@ucsf.edu)

Next-generation sequencing (NGS) is being used in numerous ways in both basic and clinical research. At the core of many studies is the human reference genome, which provides a genomic scaffold for read alignment and downstream identification of genomic variants[1–3]. However, despite the tremendous sequencing effort and methodological advances, the creation of a comprehensive human reference genome set that can represent the genetic variations across populations is yet to be realized. While the current human reference genome assembly (hg38. p11) is the most complete version to date, only 261 alternate loci are included to provide representation of haplotype diversity[4]. The remainder of the genome is still being represented as a single consensus haplotype. Many lines of evidence suggest that there are unique insertional sequences not currently represented in the reference genome[3,5–7]. We use the term non-reference unique insertions (NUIs) to describe unique sequences that are found in other individuals but not in the human reference genome. Specifically, NUIs are full-length insertions that harbor at least 50 bp of non-repetitive sequences not found in the hg38 reference set, including alternative haplotypes and patches. Translocation events are also excluded from the dataset. The exact definition and selection procedures used in this paper are described in Methods.

Accumulating evidence has shown that structural variations (SVs) contribute significantly to genome diversity but SVs are not identified in routine NGS[3,7–10]. While numerous deletion algorithms can pinpoint deletion breakpoints with high sensitivity because they can be detected in a single short read, it is challenging to detect and localize long insertions comprehensively because whole genome de novo assembly is required. Several studies have identified long, non-reference insertions and shed new light on the complexity of the human genome. The 1000 Genomes Project (1000GP) structural variation consortium discovered 128 non-reference insertions in their pilot phase SV release set[11]. Other SV detection studies like the Genome of the Netherlands (GoNL) project[3], the Simon Genome Diversity project[12], and the deep sequencing of 10,000 individuals[6] have, respectively, found 7718, 950, and 4876 genomic segments not mapped to the human reference genomes. However, no breakpoint information was provided by any of these groups. Recently, deCODE genetics/Amgen discovered 3791 breakpoint-resolved non-reference sequences from 15,219 Icelandic individuals[5]. Despite the large sample size, this study involves a homogenous population and does not capture the global genetic variation. Furthermore, all the published studies exclusively used Illumina WGS with <45X coverage, without any long-range sequence information necessary to bridge repetitive elements for accurate NUI placement.

In this study, we analyze in silico phased (haplotype-resolved), de novo human genome assemblies generated with the 10×G "Linked-Read" technology. Using a custom pipeline, we identify 1842 breakpoint-resolved NUIs in 17 1000GP individuals originating from 5 different super-populations. Also, we find that these NUIs follow a population-specific pattern, which is consistent with the previous studies using single nucleotide polymorphisms[13,14]. Over half of the NUIs are also found in non-human primate genomes, suggesting that they are ancestral to humans. Furthermore, some NUIs align uniquely to entries in the Expressed Sequence Tag database (dbEST)[15] or reads from RNA sequencing (RNA-Seq) experiments, indicating that they are part of the transcriptome. Our results underline the need of a set of human reference genomes that fully incorporates the diversity of sequences across populations as sequencing of individuals across the world becomes routine.

## Results

### Genome assemblies of 17 individuals from 5 populations.
Based on the 1000GP, 14 individuals representing populations most distinctive from one another were selected for 10×G WGS using "Linked-Read" technology. We additionally downloaded 10×G WGS data of 3 other 1000GP samples from the 10×G website. This dataset includes 5 Africans (AFR), 3 Americans (AMR), 4 East Asians (EAS), 3 Europeans (EUR), and 2 South Asians (SAS). All samples were sequenced to ~60X mean read depth (except for HG00733 and NA19240, that were sequenced to 79X and 89X, respectively) with a median molecule length of 103 kb. Sequencing reads were aligned to the hg38 human reference genome using the software Long Ranger. De novo assemblies of these samples were also generated using Supernova to yield diploid pseudo-haplotypes. The average scaffold and phased block N50s for these assemblies are 18 and 3 Mb, respectively. Full summary statistics of the assemblies are found in Supplementary Table 1.

### NUI discovery pipeline and validation.
An alignment-based and de novo assembly-based custom pipeline was built to identify NUIs. The pipeline first extracted high-quality sequence reads that did not align well to the human reference genome hg38 (Supplementary Fig. 1 and Methods). These reads were aligned to the individual's diploid de novo assembly and the regions containing clusters of the reads were identified. The corresponding assembled sequences and their flanking regions were extracted and realigned to the reference genome to compute the exact breakpoints. NUIs aligning to any alternative haplotypes or patches were removed from downstream analysis. NUIs from all 17 samples were merged to generate a unified, non-redundant call set. To investigate whether the NUI count per individual was sensitive to the quality of the sequencing reads and the sequencing depth, we computed the Pearson correlations between these parameters. The Pearson correlations ($r$) were 0.37 ($p = 0.14$; Supplementary Fig. 2a) and 0.26 ($p = 0.32$; Supplementary Fig. 2b), respectively, indicating that there was no such evidence for technical bias.

To validate our call set, we used an orthogonal approach to detect insertional sequences. We generated optical genome maps with fluorescent labels marking Nt.BspQI nicking endonuclease recognition sites. SVs could be inferred by comparing the distances between two adjacent labels. Due to the inherent constraints of optical mapping, this strategy only allowed us to detect large insertions (>2 kb in size). We applied two SV calling pipelines: one from BioNano Genomics and one from a modified version of OMSV[16]. Detailed methods used to validate the NUI call set are described in Methods. Of all the NUIs over 2 kb in length, the average precision rate is 88.4% (Supplementary Table 2), corresponding to an average of 61 validated insertions out of 69 called NUIs per sample. Most of the discordant NUIs are between 2 and 3 kb in size, a size range where the BioNano SV calling algorithm is known to have a higher false positive rate due to sizing errors, especially in regions with sparse label density.

### The structure of genetic diversity across populations.
Overall, the unified, non-redundant NUI call set includes 1842 variants. They add up to 2.1 Mb genomic sequences not found in the human reference genome or the alternative haplotypes and patches (Table 1; Supplementary Data 1). Each individual has an average of 690 NUIs (Table 1; Supplementary Data 2) that represent 711 kb of so far undescribed genomic content. Of the NUIs identified, 32% are shared across all five populations while 5% are found in all 17 individuals. The NUIs are found on all human chromosomes (Fig. 1a), with 25% are <131 bp in size, half are <450 bp, and 75% are <1260 bp (Fig. 1b).

As expected, Africans have the most NUIs while Europeans have the fewest (Fig. 1c; ANOVA $F(4,12) = 5.643$, $p = 0.0086$ followed by Tukey [AFR-EAS] $p = 0.0389033$; [AFR-EUR] $p = 0.0067794$). Significant difference in NUI count is observed

**Table 1 Non-reference unique insertions summary**

| Sample | Super population | NUI count | Total bp | Median NUI count in each population |
|---|---|---|---|---|
| HG02623 | AFR | 747 | 751,291 | 747 |
| HG03115 | | 727 | 742,200 | |
| NA19240 | | 762 | 777,935 | |
| NA19440 | | 784 | 829,736 | |
| NA19921 | | 703 | 763,748 | |
| HG00733 | AMR | 657 | 687,094 | 670 |
| HG01971 | | 670 | 677,269 | |
| NA19789 | | 709 | 724,723 | |
| HG00512 | EAS | 648 | 651,655 | 665.5 |
| HG00851 | | 667 | 692,452 | |
| NA18552 | | 708 | 726,857 | |
| NA19068 | | 664 | 705,625 | |
| HG00250 | EUR | 639 | 652,081 | 639 |
| HG00353 | | 663 | 696,507 | |
| NA20587 | | 621 | 622,096 | |
| HG03838 | SAS | 635 | 662,632 | 682.5 |
| NA21125 | | 730 | 728,111 | |
| Total (non-redundant) | | 1842 | 2,107,893 | |

between Africans and East Asians but this difference is much more striking between Africans and Europeans. Principal component analysis (PCA) of NUIs shows a population-specific pattern (Fig. 1d). Specifically, PC1 clusters African samples away from other populations, while PC2 separates the East Asians from the Europeans, the South Asians, and the Americans.

**NUI origin.** To determine the origin of the NUIs, we aligned them to four different non-human primate genomes: chimpanzee (*Pan troglodytes*), gorilla (*Gorilla gorilla*), orangutan (*Pongo pygmaeus*), and bonobo (*Pan paniscus*). Over half of the NUIs aligned to the chimpanzee (1059; 57%), the gorilla (1017; 55%), and the bonobo (916; 50%) genomes (Table 2). Just a quarter of the NUIs aligned to the orangutan genome (498; 27%). This trend correlates well with the evolutionary divergence times inferred based on mutational profiling[17–19]. In aggregate, 1175 (64%) NUIs can be aligned to at least one primate genome and 451 (24%) are present in all four non-human primate genomes (Fig. 2a). The large number of ancestral sequences implies that the donors of the human reference genome came from human-specific lineages where these sequences were deleted after the human–chimpanzee split.

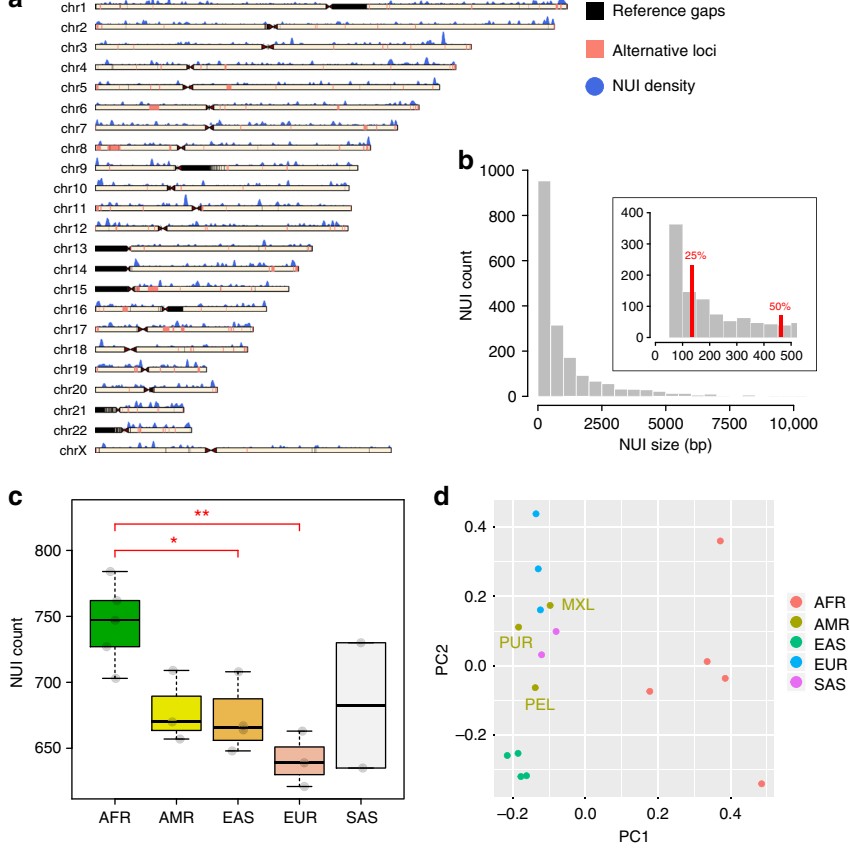

**Fig. 1** Overview of the non-reference unique insertions and their distributions across populations. **a** Ideogram depicting NUI occurrences across all chromosomes. The blue histogram above each chromosome describes the number of NUI using a sliding window of 1 Mb with a 10 kb step size. Alternative loci incorporated in hg38.p11 are shown in pink, and for display purpose, the sizes of these loci are extended by 100 kb on both sides. **b** NUI size distribution using a bin size of 500 bp in the main plot and 50 bp in the zoomed area. The 25th and 50th percentiles are labeled in red. NUIs longer than 10 kb in length were removed before plotting. **c** The number of NUIs across all five super-populations. Each gray dot shows the actual NUI number per individual. ANOVA $F(4,12) = 5.643$, $p = 0.0086$ followed by Tukey [AFR-EAS] $p = 0.0389033$; [AFR-EUR] $p = 0.0067794$. The box plot illustrates the median, the upper and lower quartiles for each population. Since no points exceed the 1.5 X interquartile range, the whiskers correspond to the minimum and maximum values in each group. (\*$p \leq 0.05$; \*\*$p \leq 0.01$). **d** The first two principal components based on the NUI occurrence matrix. The sub-populations of the American samples were labeled on the plot. AFR Africans, AMR Americans, EAS East Asians, EUR Europeans, SAS South Asians. American sub-populations: MXL Mexican Ancestry in Los Angeles, CA, USA; PEL Peruvian in Lima, Peru; PUR Puerto Rican in Puerto Rico

**Table 2 Non-reference unique insertions identification in non-human primates**

|  | Number of aligned sequences | Aligned sequence percentage | Number of sequences found in all 5 populations | Number of sequences found in all 17 individuals |
|---|---|---|---|---|
| Chimpanzee (panTro5) | 1059 | 57% | 475 | 85 |
| Gorilla (gorGor5) | 1017 | 55% | 445 | 85 |
| Bonobo (panPan2) | 916 | 50% | 407 | 78 |
| Orangutan (ponAbe2) | 498 | 27% | 229 | 51 |
| Union of all non-human primates | 1175 | 64% | 509 | 91 |
| Total sequences in the dataset | 1842 | 100% | 584 | 95 |

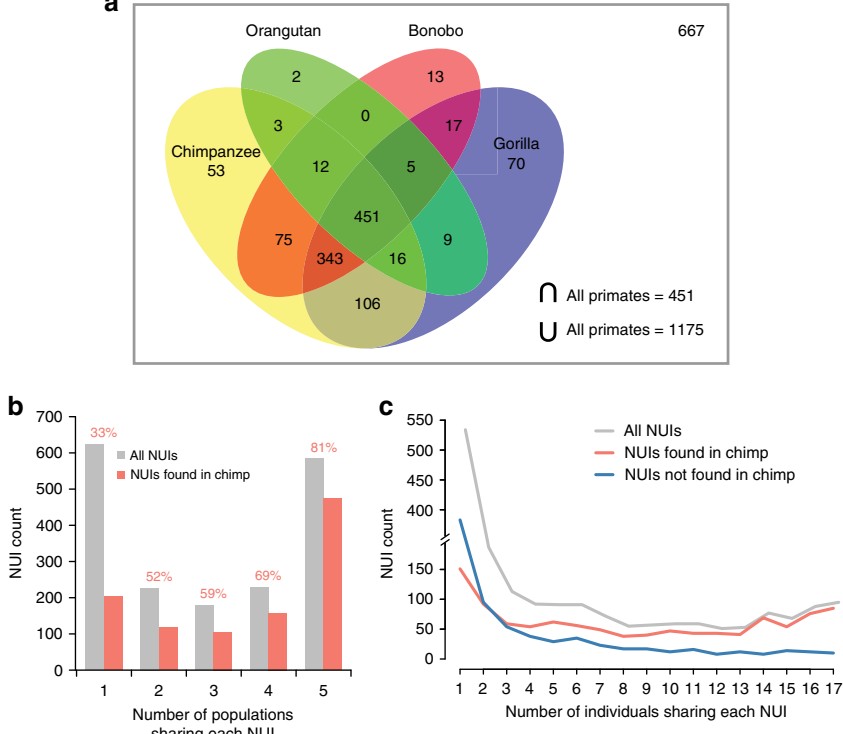

**Fig. 2** Non-reference unique insertions in non-human primates. **a** Venn diagram illustrating the number of NUIs found in non-human primate genomes. The gray box represents the universe, which includes 658 NUIs that were not found in any non-human primate genomes. **b** NUI frequency stratified based on the number of populations sharing these sequences. Frequencies are further broken down depending on whether they aligned to the chimpanzee genome. The numbers on top of the bars represent the percentages of NUIs identified in the chimpanzee genome ($\chi^2 = 309.5$, $p = 9.71e-66$). **c** As in **b** but at the individual-level

Next, we assessed whether the more common NUIs were enriched for ancestral sequences. Our analysis shows that the NUIs shared across all five human populations are significantly enriched in the chimpanzee genome (Fig. 2b; $\chi^2 = 310.31$, $p = 6.46e-66$). Remarkably, 81% of the NUIs found in all five human populations are also present in chimpanzee genome whereas only 33% of the NUIs found in a single population are present in the chimpanzee. A similar trend is observed at the individual level, where NUIs found in at least four individuals are much more likely to be found in the chimpanzee genome (Fig. 2c).

**Analysis of repeat sequences associated with NUIs**. We ran RepeatMasker[20,21] on the entire NUI call set to analyze the composition of transposable elements (TEs) (Supplementary Table 3). We found that 21.4% and 23.4%, respectively, of the overall NUIs were short interspersed nuclear elements (SINEs) and long

interspersed nuclear elements (LINEs). In contrast to the genome-wide repeat content of SINEs and LINEs (13% and 21%)[22], only SINEs were significantly enriched in this dataset. To explore the distributions of TEs, we sorted the major types of TEs into NUIs of difference sizes (Fig. 3a). NUIs under 200 bp are mostly unique sequences while longer NUIs associate mostly with SINEs, especially *Alu* elements. LINEs are the next most abundant TE found in NUIs longer than 200 bp. Long terminal repeats (LTRs) and DNA transposons are found at lower frequency in all size ranges. We additionally characterize the flanking repetitive sequences to identify potential mechanisms mediating the formation of NUIs. We found that 63% of the NUIs were flanked by a TE on at least one end (Table 3).

When aligning NUIs to the human reference genome, we observed that the two ends of the NUIs occasionally contained homologous sequences that collapsed into one overlapping copy in the reference sequence (Fig. 3b). This type of NUI account for

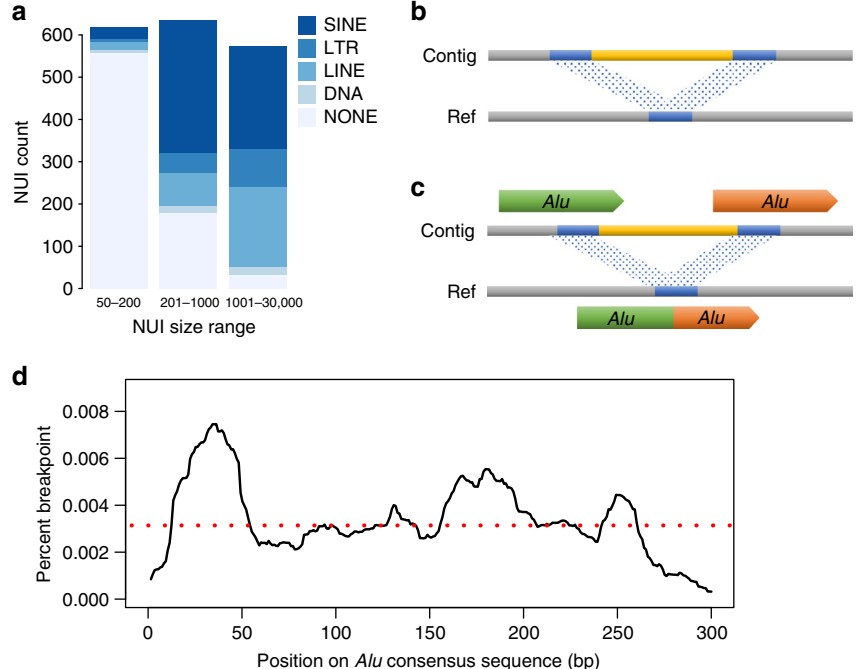

**Fig. 3** Characteristics of non-reference unique insertions. **a** Major transposable element in each NUI split by three different size ranges. Stacked bars represent the total number of NUI in each size range. SINE short interspersed nuclear element, LTR long terminal repeat, LINE long interspersed nuclear element, DNA DNA transposon, NONE no interspersed repeat detected. **b** Example of an overlapping alignment in the reference where both ends of the NUI are homologous. Yellow block, NUI; blue blocks, homologous sequences; contig, a stretch of sequence containing the NUI; ref, reference. **c** When two *Alus* recombine ectopically, the resulting alignment, as shown in the reference strand, becomes a chimeric *Alu*. **d** Percentage of breakpoint along an *Alu* consensus sequence. The red dashed line represents the average breakpoint percentage (0.0033%)

**Table 3 Distribution of transposable elements flanking all non-reference unique insertions**

|  | Count | Percent |
|---|---|---|
| Alu/Alu | 336 | 18% |
| Alu/LINE | 49 | 3% |
| LINE/LINE | 105 | 6% |
| Other TE | 673 | 37% |
| Non-TE | 679 | 37% |
| Total NUIs | 1842 | 100% |

**Table 4 Distribution of transposable elements flanking non-reference unique insertions with at least 10 bp homologous sequences on both ends**

|  | Count | Percent | Found in chimp | Percent in chimp |
|---|---|---|---|---|
| Alu/Alu | 265 | 52% | 163 | 62% |
| Alu/LINE | 5 | 1% | 4 | 80% |
| LINE/LINE | 13 | 3% | 4 | 31% |
| Other TE | 135 | 26% | 40 | 30% |
| Non-TE | 92 | 18% | 25 | 27% |
| Total NUIs | 510 | 100% | 236 | 46% |

at least 28% of the entire call set (510 out of 1842 NUIs have at least 10 bp overlap). Shorter overlaps are also observed but 10 bp is used as a threshold for our analysis. Using RepeatMasker on this NUI subset, we identified 418 (82% of this subset) NUIs flanked by a TE on at least one end (Table 4), with 52% flanked by *Alu* elements on both ends.

This observation, where two *Alu* elements flank many NUIs on both ends, suggests that *Alu* recombination-mediated deletion (ARMD)[23,24] is responsible for their formation. Specifically, recombination between two different *Alu* elements not in equivalent positions can give rise to ARMD and leave behind a single *Alu* chimera (Fig. 3c). In other words, the deleted version is found in the reference genome while the NUIs represent the ancestral sequences. Of the 265 candidate ARMDs identified in this dataset, 163, 167, 102, and 38, respectively, are also identified in the genomes of the chimpanzee, gorilla, bonobo, and orangutan (Table 4; Supplementary Fig. 3). The union of the ARMDs found across the four non-human primates yield a total of 195 events, which is equivalent to 74% of the total candidate ARMDs found in the NUIs. Among those, *AluSx* and *AluY* are the two most abundant *Alu* sub-species responsible for these

events. This observation is congruent with the genome-wide copy number of these two *Alu* sub-family in the primate genome[25]. Moreover, we identified two putative recombination hotspots at the 12–52 and 156–205 bp positions of *Alu* elements (Fig. 3d). The first recombination hotspot is in accordance with a previously published report[23], bolstering the idea that there might be short highly conserved sequences that allow for frequent ARMDs.

**Transcription potential of NUIs.** Although 99% of the NUIs are located in the intergenic and intronic regions of the genome, there is a possibility that they are previously unannotated exons or regulatory elements with transcription potential. We used two orthogonal approaches to determine whether any of the NUIs were transcribed. First, we aligned the NUIs to the human Expressed Sequence Tag database (dbEST) and found 129 NUIs (7%) uniquely aligning to the human dbEST.

To extend this line of analysis, we also aligned the NUI call set to the high-quality RNA-Seq reads from the Geuvadis project[26].

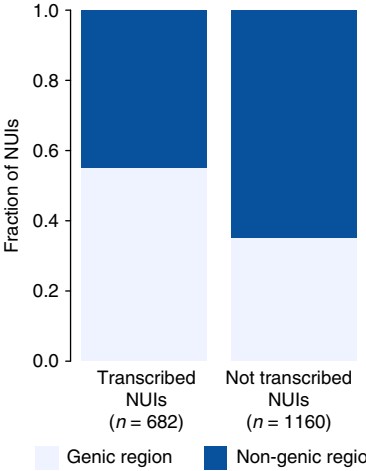

**Fig. 4** Characterization of non-reference unique insertions in the transcriptome. Stacked bars representing the proportions of NUIs that are either in the genic or non-genic regions of the genome (Fisher $p = 8.12e-17$, OR = 2.26)

We randomly selected 50 individuals representing two super-populations (Europeans and Africans) and extracted RNA-Seq reads not mapped to the human reference genome. We realigned the unmapped RNA-Seq reads to our dataset to identify NUI bearing transcribed sequences (Methods). From the Geuvadis RNA-Seq data, we identified 643 trancribed NUIs (35%), among which 90 overlapped with the result from the human dbEST. This overlap of transcribed NUIs is statistically significant (Fisher's $p = 8.018409e-17$; OR = 4.83). In total, 682 NUIs are transcribed and more than half are located in the genic region of the human genome (Fig. 4; Fisher $p = 8.12e-17$, OR = 2.26).

**Comparison with the Icelandic deCODE Genetics dataset.** Recently, deCODE Genetics/Amgen analyzed 15,219 Icelanders to identify a set of 3791 breakpoint-resolved non-reference sequences not found in the human reference genome[5]. To compare our dataset with the published results, we aligned all the NUIs to the sequences reported by deCODE Genetics in a reciprocal manner (Methods). We found that 578 of our 1842 NUIs (31%) intersected with the deCODE dataset. After filtering out the singleton NUIs, 1308 remain in our dataset and 516 (39%) are also found by deCODE Genetics. Of these 516 NUIs, 258 (50%) are present across all five populations and 463 (90%) are found in non-human primates. Given the fact that Icelanders are of British and Norwegian origin, we then assessed whether the Europeans in our study cohort shared more NUIs with this deCODE dataset than other populations. We found that among the shared NUIs, Africans had the most shared NUIs with the deCODE dataset while Europeans had the fewest (ANOVA $F(4,12) = 6.216$; $p = 0.006$; followed by Tukey [AFR-AMR] $p = 0.0349966$; [AFR-EAS] $p = 0.0462222$; [AFR-EUR] $p = 0.0058295$). This observation suggests that most of the shared NUIs are not European-specific but are mostly ancestral sequences. Additionally, the size distributions of the non-reference sequences identified by deCODE Genetics are smaller than the ones in our call set, with only 238 (6%) of the deCODE non-reference sequences >2 kb and 29 (0.8%) > 5 kb. In contrast, our call set consists of 310 (17%) NUIs > 2 kb and 73 (4%) > 5 kb.

## Discussion

In this study, we identified a large set of NUIs using phased, high-quality de novo human genome assemblies from 17 individuals with diverse genetic backgrounds. This is, to our knowledge, the most

diverse set of high-quality whole genome assemblies to date using technologies that can infer long-range sequence information. In addition, the reconstruction of NUIs in this study confirms the highly variable population patterns of genomic inserts. The high abundance of NUIs found in Africans is not surprising as it is well-known that Africans have the highest level of genetic diversity among populations[27-30]. Europeans have the fewest NUIs, which is presumably due to the fact that about 70% of the human reference genome sequences came from a single donor of likely African–European admixed ancestry[4]. Despite having a small sample size for principal component analysis, the clustering pattern accurately recapitulates previously established population pattern and admixture. The fact that most of the NUIs are found across two or more populations indicates that the human reference genome has the minor alleles at these loci. The inclusion of minor alleles in the reference can interfere with every stage of variant discovery and downstream annotation[31-33]. This is particularly problematic as NUIs usually involve long stretches of DNA that differ from the reference. These missing sequences can hamper efficient read mapping. By integrating NUIs to the human reference genome, a more complete genomic scaffold can be used to annotate biological data.

Our observation that a significant subset of NUIs is flanked by *Alu* repeats and other TEs suggests that ARMD may be the mechanism by which they are deleted from the reference genome. *Alu* repeats and other TEs are among the major driving forces of genomic variation and evolution[34,35]. The over-representation of SINEs in this NUI dataset maybe attributable to the high frequency of *Alu* elements involved in ARMD, which accounts for 14% of the NUIs in this dataset. This mechanism has been described previously in the context of chimpanzee versus the human genomes[23,24]. However, this is the first time such ARMDs are assessed genome-wide within the human population. Here, we have shown that ARMDs can be polymorphic between individuals or populations, and some events classified as "human-specific" are in fact highly variable even within the human populations.

In identifying transcription potential of NUIs, we found a subset of sequences aligning to either the dbEST or reads from RNA-Seq experiments, suggesting that some of the NUIs contribute directly to transcriptomic diversity that has not been accounted for in the past. These sequences may include previously unannotated exons or non-coding regulatory sequences that can alter the rate of transcription of targeted genes. The inclusion of these NUIs is essential to our understanding of the human transcriptome and the transcriptional regulation of non-coding sequences. It is also important to note that only RNA-Seq from two populations were used in the analysis. The number of transcribed NUIs would certainly increase when data from more individuals or additional populations are also analyzed.

Finally, when we evaluated the non-reference sequences reported by deCODE Genetics/Amgen[5], we found that 39% of the more common NUIs were also present in Icelanders. Even with the singletons removed, 516 NUIs (89% of the initial 578 shared NUIs) still remained. This result suggests that the sequences reported by deCODE Genetics are not exclusive to Icelanders. However, since the genome of the Icelanders is relatively homogenous[36], we expect to see a small proportion of genomic alterations that are unique to them due to an elevated level of genetic drift and the potential for founder effect[37].

Overall, our diverse dataset allows us to perform a cross-population survey of NUIs that are typically difficult to detect when one relies only on short-read sequencing data, thereby expanding the NUI catalog and sequences specific to populations that are often underrepresented. The current human reference genome represents a single composite haplotype at any given locus. In this linear form, and even with the inclusion of a limited set of alternate haplotypes, it cannot capture much of the genetic

diversity across the different continental groups. Several studies have already demonstrated that the use of an ethnicity-matched reference allows for increased accuracy in imputation, and ultimately, disease susceptibility prediction[38–41]. It is therefore critical to produce a set of human reference genomes that includes NUIs from many populations in order to depict the unbiased landscape and diversity of the human genome.

## Methods

**Collection of 10×G Linked-Read data.** Linked-Read data for 14 samples were generated on site using cell lines purchased from the Coriell Institute. Another three samples, namely HG00512, HG00733, and NA19240, were obtained from the 10×G website in the format of FASTA and FASTQ files (https://support.10xgenomics.com/de-novo-assembly/datasets/1.1.0/msHG00512; https://support.10xgenomics.com/de-novo-assembly/datasets/1.1.0/msHG00733; https://support.10xgenomics.com/de-novo-assembly/datasets/1.1.0/msNA19240). The FASTA files corresponded to the pseudo-haplotypes that were generated with Supernova v1.1 while the FASTQ files were downloaded so that we could generate the alignment BAM files using Long Ranger v2.1 in-house.

**Collection of unaligned/poorly aligned read pairs.** The NUI discovery pipeline initially accepted a BAM file generated from 10×G Long Ranger v2.1 and identified reads that did not align well to the human reference genome (core hg38). This was defined as reads fulfilling at least one of the following criteria:

- Reads with an unaligned SAM flag, and their mates
- Read pairs not mapped within insert sizes (BWA-MEM[42] estimated the insert size based on the bulk read pair distributions and this information was used by the Long Ranger alignment software)
- Read pairs with wrong read orientations
- Reads with an alignment score $\leq -80$, and their mates (this corresponds to a minimum of 40 mismatches or a combination of other penalties according to the Lariat scoring parameters: AS:f $= -2 *$ mismatches $-3 *$ indels $-5 *$ clipped $-0.5 *$ clipped_bases $-4 *$ improper_pair)
- Reads with more than 40 bases clipped off, and their mates

Samblaster v0.1.24[43] was used to extract reads with clipped and unaligned reads while sambamba v0.5.9[44] and samtools v1.2[45] were used to collect other poorly aligned reads.

The raw FASTQs of this collection of unaligned/poorly aligned read pairs were extracted using seqtk v.1.0[46] and processed by trimming off the first 23 bp of the first mate of each read pair to remove the 16 bp 10×G barcode and the 7 bp low accuracy sequence from an N-mer oligo. This collection of reads was then filtered based on their base qualities using the fastq_quality_filter utility provided in the FASTX Toolkit v0.0.14[47]. The entire reads, along with their mates, were removed if less than 70% of their bases had a quality score of 30 or above.

**Alignment of unmapped reads to 10×G pseudo-haplotypes.** BWA-MEM[42] paired-end mode was used initially to align all the unmapped/poorly mapped read pairs to the hg38 reference genome and the sample-matched pseudo-haplotypes generated from 10×G Supernova v1.1. Starting from this alignment step onward, all the procedures were repeated for both pseudo-haplotypes. After read alignment, read pairs that mapped well to the hg38 reference genome were discarded from the other two alignment outputs, which were further filtered based on the following stringent criteria:

- Read1 with an alignment score of at least 90
- Read2 with an alignment score of at least 113 (read1 was shorter due to trimming of the barcode and the N-mer oligo)
- Read pairs in the proper orientation
- Read pairs mapped within insert size
- Mapping quality was at least 30

The alignment scores used in this analysis (90 and 113) were determined based on the BWA-MEM alignment scoring scheme. Each mismatch gets assigned a penalty of $-4$, and hence 90 and 113 roughly correspond to 9 mismatches in the sequence alignment (or a combination of penalties).

**Identification of read clusters.** Keeping only reads that aligned well to the pseudo-haplotypes, the read coverage at every position was calculated using bedtools[48]. Read clusters with coverages between 8 and 100 were located. The pseudo-haplotype sequences corresponding to the read clusters were extended by 7000 bp on each end, or until the ends of the assembled sequences, to serve as the alignment anchors for the following step. If the ends between two different contigs were overlapping or separated by less than 200 bp, the entire sequence from the upstream left anchor to the downstream right anchor was extracted for Lastz[49] alignment. Contigs with more than 10 N's were removed from downstream analysis to ensure high accuracy.

**Breakpoint computation.** Extended contigs were aligned to the hg38 core reference genome using Lastz with the following parameters: –step $= 20$ –seed $=$ match15 –notransition –exact $= 400$ –identity $= 99$ –match $= 1,5$. We then computed the precise breakpoint in each contig by locating where the sequence alignment broke off and realigned. Ideally, one contig should produce two alignments as each anchor should align uniquely to the reference genome, separated by one or two breakpoints. This is, however, not always the case. When part of an anchor aligned to multiple places on the reference, we selected the one with the longest alignment and the highest sequence identity. If an anchor aligned to more than five genomic loci, we ensured that the anchor alignment was at least 3500 bp in length before choosing the best alignment candidate. When the two anchors from the same contig partially overlapped on the reference, we filtered them out if the overlapping sequence was larger than 800 bp in size to ensure the accuracy of our call set. Alignments that created an overlap of 800 bp had poor concordance rate with the BioNano insertion calls. Even when BioNano makes an insertion call at the genomic locus overlapping an NUI candidate, the median size difference between BioNano-predicted size and the length of the NUI was about 3 kb. In contrast, the median size difference for all other alignments were usually between 300 and 600 bp. While BioNano cannot determine the precise insertion size due to its inherent resolution limits, this difference suggested that the sequences with larger overlap were not as reliable. To ensure the high quality of our call set, these sequences were discarded prior to the analysis. At this point, only insertional sequences with gap size $\geq 50$ bp were kept for downstream analysis. Contigs whose ends were immediately flanked by N-gaps were discarded. Output from the two pseudo-haplotypes were combined at this point. A unified list was generated by combining homozygous contigs, that was, if their breakpoints were less than 10 bp apart on each side. The unaligned breakpoint-to-breakpoint sequences of all of the contigs were extracted for further analysis.

**Definition of NUI variants.** To determine whether these contigs fulfilled the definition of an NUI, we ran RepeatMasker v4.0.7 and dustmasker[50] to determine the extent of interspersed repeats and low complexity sequences in each contig. RepeatMasker was run with -species human and dustmasker was run with the default settings. The number of unmasked base pairs was counted for each contig and sequences with $\geq 50$ unique bases were kept. To ensure that these contigs were not included in the human reference genome including all the alternate haplotypes, fix patches, and novel patches, we used BLAST to align these NUIs—extended by 50 bp on both ends—to the hg38.p11 human reference genome. NUIs aligning to hg38.p11 with $\geq 95\%$ identity and 100% coverage were removed from the call set. We also removed NUIs resulting from translocation events. To identify translocated sequences, we used BLAST to align NUIs—breakpoint-to-breakpoint—to the human reference genome with –task megablast and –dust no. Alignments with $\geq 95\%$ identity and 100% coverage were removed from the call set. To reduce false-positive calls, we filtered out NUIs whose breakpoints overlap assembly gaps, segmental duplications, and other problematic regions as defined by the 10×G SV filter criteria: https://support.10xgenomics.com/genome-exome/software/pipelines/latest/advanced/sv-blacklist. Two files: sv_blacklist.bed and segdups.bedpe were used for this filtering step. Finally, we merged all the NUIs across 17 individuals to make a unified, non-redundant call set by collapsing NUIs if the inserted sequences shared one identical breakpoint with another sequence or if both breakpoints (start and end) were within 50 bp of those from another sequence. The sequences in FASTA format and the NUI occurrence matrix can be found in Supplementary Data 1 and 2, respectively. The reported NUI occurrence matrix is encoded as 0, 1, and 2 (2 meaning the individual harbors the NUI in both pseudo-haplotypes). However, since the homozygous calls were not definitive, all the NUIs with a genotype of 2 were recoded to 1 to ensure high data quality. The recoded binary matrix was used for the rest of the analysis. We believe that some of the homozygous calls were incorrect due to the observations that many of the NUI singletons are homozygous calls rather than heterozygous calls. In these cases, the Supernova assembler places the NUIs in both pseudo-haplotypes inappropriately.

**NUI validation using BioNano optical maps.** We used BioNano optical maps insertion calls to validate our NUI call set. Our SV calling strategies involved two pipelines: BioNano pipeline 4618/4555 and a modified version of OMSV[16]. Insertions called by either algorithm are merged together for downstream analysis. To validate, we first identified a list of NUIs that were greater than 2 kb in length. SVs smaller than 2 kb are prone to sizing error caused by DNA fragments that are either not completely linearized or over-stretched.

Next, we overlapped NUIs with the individual-matched optical maps and filtered out insertions where the optical maps had zero-coverage, or if they were within 10 kb of these zero-coverage regions. Finally, we filtered out NUIs if another SV was reported within 10 kb, including deletions, complex inversions, or simple insertions that did not fulfill the definitions of NUIs. Exclusion of these NUIs were necessary since the BioNano optical maps only show the overall size change between two labels. In other words, BioNano SV calling is error-prone if multiple SVs occur in tandem.

Taking this set of filtered NUIs, we computed the precision rate for each individual. The precision rate was calculated as follows:

$$\text{precision} = \frac{\text{NUIs supported by BioNano}}{\text{NUIs supported by BioNano} + \text{NUIs not supported by BioNano}}$$

**Contamination screen**. We did not expect sequencing contaminants from bacteria, virus, plants, and yeast to be present in this NUI call set because they should not have strong anchor to the human reference genome. Additionally, contaminants would not have significant barcode sharing with nearby sequences, and hence they would not form long contigs. To verify these assumptions, we used BLAST to align NUIs to the bacteria database (ftp://ftp.ncbi.nlm.nih.gov/genomes/genbank/bacteria/), the human microbiome database (ftp://ftp.ncbi.nih.gov/genomes/HUMAN_MICROBIOM/Bacteria/all.fna.tar.gz), the univec database (ftp://ftp.ncbi.nlm.nih.gov/pub/UniVec/UniVec), and the virus database (ftp://ftp.ncbi.nih.gov/genomes/Viruses/all.fna.tar.gz). None of the NUIs produced significant alignment with these common contaminants using 90% identity and 90% query coverage filter thresholds.

**Principal component analysis**. The NUI occurrence matrix was used as the input for the principal component analysis. To clean up the matrix, NUIs with 1, 2, and 17 occurrences were removed from the matrix before analysis. NUIs with 1 or 2 occurrences might increase ambiguity in the dataset while NUIs with 17 occurrences (found in every individual) had no variance. Overall, 1026 NUIs remained in the matrix for analysis.

**Aligning NUIs to non-human primates**. Four non-human primates were used in this study and their reference genomes were downloaded from either UCSC or NCBI. The specific versions used in this study were chimpanzee (panTro5), gorilla (gorGor5), bonobo (panPan2), and orangutan (ponAbe2). We used BLAST to align the NUIs, including 50 bp flanking sequences, against each of these non-human primate genomes using -task megablast and -dust no. Only sequences that aligned with at least 95% identity and 95% query coverage were considered as real hits.

**Identifying the major transposable element in NUIs**. To identify the major TE in each NUI, we ran RepeatMasker individually to compute the sequence composition for each TE. For each entry, the TE that makes up the highest percentage of that sequence was deemed the major TE. RepeatMasker was run with –species human, -xsmall, and -nolow.

**Determining transposable elements flanking NUIs**. To determine whether the two ends of the NUIs were flanked by TE, we extracted 300 bp upstream and 300 bp downstream of each end and ran RepeatMasker to determine the composition of these sequences. We specifically filtered for TEs containing the NUI breakpoints to determine the potential mechanisms mediating these insertions.

**Identifying breakpoint frequency in an *Alu* element**. We assessed the breakpoint frequency along an *Alu* element consensus sequence to identify potential hotspots for the ARMD. First, we identified *Alu*-flanking NUIs that shared homologous sequences on both ends. These sequences form overlapping alignments on the reference and 10 bp sequence homology was required. Based on the RepeatMasker output from before, we determined the position of the *Alu* element corresponding to the breakpoint, and we subtracted the length of overlap from that breakpoint position to obtain a breakpoint range for that particular NUI. Since the breakpoints are ambiguous over a range of positions, only $\frac{1}{\text{length of range}}$ was added per each position to compute the breakpoint frequency.

**ARMD size distributions**. Similar to the last section, we first extracted all the NUIs flanked by *Alus* on both ends and we further filtered to keep the ones where the two *Alus* came together to form a single *Alu* chimera on the reference. This final set of NUIs was used to create Supplementary Fig. 2.

**Aligning NUIs to the expressed sequence tag database**. We used BLAST to align the NUIs to the human dbEST (ftp://ftp.ncbi.nlm.nih.gov/blast/db/est_human.[number].tar.gz). Regions with 95% sequence identity, regardless of the query coverage, were extracted for further assessment. We adjusted the filter criteria here because entries in the dbEST are short and the sequences usually only represent the ends of expressed genes. Next, the regions that aligned to the dbEST were realigned to hg38 with BLAST, to ensure these regions do not align to anywhere else on the genome. Any of these regions that aligned to the reference genome were discarded. Otherwise, these sequences were considered to be transcribed.

**Aligning RNA sequencing reads to the NUI call set**. We gathered RNA-sequencing reads from the Geuvadis project[26] and the raw sequencing data is publicly available. We randomly selected 50 individuals for the analysis—10 from each sub-population included in the Geuvadis project (CEU-EUR, FIN-EUR, GBR-EUR, TSI-EUR, and YRI-AFR). Raw FASTQ files for the RNA-Seq reads were downloaded from https://www.ebi.ac.uk/arrayexpress/experiments/E-GEUV-3/. To align RNA-Seq reads, we first generated the genome index using the hg38 primary reference genome and the GTF file downloaded from the ensembl website (ftp://ftp.ensembl.org/pub/release-92/gtf/homo_sapiens/Homo_sapiens.GRCh38.91.gtf.gz) with the following parameters: STAR –runThreadN 32 –runMode genomeGenerate –genomeDir /path/to/STAR_genome –genomeFastaFiles /path/to/hg38/fa –s–jdbGTFfile /path/to/emsembl_gtf –sjdbOverhang 74. We aligned all RNA-Seq

reads to this indexed genome, one sample at a time, as follows: STAR –runThreadN 32 –outReadsUnmapped Fastx –genomeDir /path/to/STAR_genome –outFileNamePrefix /out/path/prefix –outFilterMultimapNmax 100000 –outSAMunmapped Within KeepPairs –limitOutSAMoneReadBytes 1000000 – readFilesIn read_1.fastq read_2.fastq. By default, STAR does not output unmapped reads in the SAM alignment file. This feature has to be turned on with –outSAMunmapped and –outReadsUnmapped. The second parameter automatically wrote all the unmapped FASTQs into separate files that could be used directly for the subsequent steps. Since STAR aligner considers reads unmapped if they are mapped to more than 20 loci by default, this feature was turned off by setting this threshold to 100000 using –outFilterMultimapNmax. We then collected all of the unmapped reads and their mates from the 50 individuals. Specifically, unmapped reads were defined by the SAM flag 4. We merged the FASTQ files from all samples into a single read1 file and a single read2 file.

Since RNA-Seq aligners behave differently when aligning to a reference with and without the GTF annotations[51], we realigned the collection of unmapped reads to the human reference genome without supplying the GTF file. All unmapped reads from this second alignment step were collected for the final alignment procedure, which involved mapping these reads to the NUI call set in which every NUI was extended by 300 bp on both ends. The NUI extension step was essential to facilitate reliable alignments since some of the NUIs were too short for mapping. The genome index for the NUI was generated using the same parameter described above, except we did not provide a GTF file. The unmapped reads were passed on to the STAR aligner using the following command: STAR –runThreadN 32 –genomeDir /path/to/ STAR_genome_NUI –outFileNamePrefix /out/path/ prefix –readFilesIn unmapped_read1.fq unmapped_read2.fq. Reads were then filtered based on the following criteria:

- Both reads in a read pair were mapped
- Both reads in a read pair could not fully reside in the 300 bp flanking sequences
- Read pairs mapped in correct orientation and with correct insert size
- Reads with alignment scores ≥ 140 (this is equivalent to four total mismatches for a read pair)
- Reads with a mapping quality score of 255, which is indicative of unique mapping based on the STAR scoring scheme

NUIs aligning to at least any one of these filtered read pairs were considered to be transcribed.

**Comparison with the deCODE genetics (Icelandic) dataset**. The non-reference sequences generated by deCODE Genetics/Amgen were obtained from supplementary data 1 of the publication[5]. We first aligned all the NUIs to the Icelandic contigs using BLAST. Alignments with 95% sequence identity and 95% query coverage were considered as real hits. Next, we reciprocally aligned the Icelandic contigs to the NUIs also with BLAST, and the alignment resu7lts were filtered using the same criterial described above. Results from the two alignment steps were merged and a non-redundant list of NUIs was used for the reported statistics. Reciprocal sequence alignment as performed since the reported breakpoints between the two pipelines might not be consistent. After the initial analysis, we also filtered out uncommon NUIs from our NUI call set. Namely, NUIs with one occurrence in the study cohort were discarded. These sequences are likely to be population-specific and we would not expect them to be shared with the Icelandic genomes.

**Code availability**. The NUI pipeline source code can be accessed via GitHub (https://github.com/wongkarenhy/NUI_pipeline.git).

**Data availability**. The 10×G read alignments (BAMs), the Supernova assemblies (FASTAs), and the BioNano assemblies (CMAPs) can be accessed via NCBI Bio-Projects PRJNA418343 and PRJNA435626. All NUIs were deposited to NCBI GenBank nucleotide database with accession numbers MH533022-MH534863. Three additional samples, HG00512, HG00733, NA19240, were obtained from the 10×Genomics website. The Geuvadis RNA sequencing data set was obtained from accession number E-GEUV-3.

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

## Acknowledgements

Research reported in this publication was supported by the National Human Genome Research Institute of the National Institutes of Health (NIH) under award R01 HG005946 to P.-Y.K. The content is solely the responsibility of the authors and does not necessarily represent the official views of the National Institutes of Health. We thank Yulia Mostovoy for her intellectual contributions to this work, Le Li for sharing the SV call set generated from the BioNano optical maps, and Ernest Lam for providing technical guidance for the optical mapping analysis.

## Author contributions

Pipeline development: K.H.Y.W. Data analysis: K.H.Y.W. and M.L.-S. Manuscript writing: K.H.Y.W., M.L.-S., and P.-Y.K.

## Additional information

**Competing interests:** The authors declare no competing interests.

