## [Peer Review File · Nature Communications]

Reviewer #1 (Remarks to the Author):

In their manuscript, Wong et al report their analysis of 17 human genomes sampled from heterogeneous group of individuals representing five populations. The authors obtained whole genome data generated with 10x chromium technology and employed it to detect, map and characterize genomic sequences that are absent from current human genome reference (GRCh38).

The knowledge on unknown segments in our genomes is very important as such segments are routinely ignored in analysis of whole genome sequencing and transcriptome data. A strong point of this study is the effort to characterize non-reference segments in ethnically divergent set of individuals. Another advantage over several previous studies is use of “linked-read” technology that improves anchoring of these segments to existing genome reference. The authors provide a great resource of 1828 breakpoint-resolved inserts, which will be of interest to many other researchers in the field of genomics.

Despite these positive points, there are downsides that need to be addressed before manuscript can be considered for publication.

Major points.

Manuscript is poorly written; it feels like reading lab journal. Large parts of it explains obvious concepts (e.g. details on how optical mapping works on page 16 or long discussion on minor alleles in genome and divergence among primate species on page 10). The size of the manuscript should be significantly shortened.

At the same time, some other parts of manuscript are incomplete or formulated ambiguously. I would, therefore, suggest that a major revision is needed to bring this interesting research to good shape.

Specific comments:

Page 3. Authors use term Novel Sequence Insertion, while they also show their presence in genomes of other primates. I suggest to use term Non-reference Sequence Insertion instead.

Page 5. References to data URLs (10G website), versions of software (Long ranger and Supernova) should be given, preferably in methods section

Pearson correlations 0.28 and 0.26 do NOT indicate that “NSI were not biased”, rather they do not show evidence for such bias.

Page 6. “average recall rate 88.2%” – you need to mention average number of validated events per sample in the main text.

“Americans are admixed with South Asians” – do you mean ‘cluster with’ here?

Page 7. “being conserved among all four non-human primates” – do you mean ‘present in genomes of all four...’

“NSI aligned to the chimpanzee genome had any underlying population structure” – too vague

“NSI ... were significantly enriched in chimpanzee” – same as above, be more specific.

“NSI sequence characterization” – too broad header for a repeatMasker analysis. Something like ‘analysis of repeat sequences in NSIs’ would better reflect the essence of that section.

Page 8. Give a reference upon first appearance of ARMD. Further you claim that NSIs are caused by ARMD, while this is a likely explanation for their origin and you do not have proof the mechanism was ARMD, e.g. “... Alu subspecies responsible for ARMD in this analysis”. Further, one of the ways to support ectopic recombination mechanism would be to check ancestral states of these loci in chimp, gorilla and bonobo.

“Biological consequences of NSI” – again too broad title when you simply check for expression evidence.

Page 9. Might be a good idea to compare Icelandic NSIs to those that you identified in European population.

Page 10. “patterns of SVs” – patterns of genomic inserts

“dominated by SEQUENCES FROM DNA donors”

“NSI could be used to distinguish the populations” – too impractical

“people of different origins” – ‘people of different ethnic backgrounds’

“Having minor alleles ... can interfere with every stage” – it is my understanding that insertions have preference over deletions for being included in genome reference. That leads to more complete genome reference.

Page 11. About half of our genomes are repeats. Did you check for significant overrepresentation of Alus/LINEs in NSIs?

“ectopic recombination ... MIGHT BE responsible for about 13% of all NSIs”

“In addition the putative recombination hotspots ...” – the discussion of this fact is bigger and properly referenced, while I would expect that in results section.

Page 13. “read pairs mapped within insert sizes” – this does not make sense to me.

Page 14. “Contigs whose ends were separated by less than 200bp are merged” – not sure what that means

Page 15. “Finally, we concatenated all the NSIs” – do you mean ‘overlapped’?

Page 17. Definition of recall rate is incorrect, formula you are giving calculates precision. What did you plan to calculate?

Page 19. When checking RNA-seq reads in NSIs – do you check that read pair does not fully reside in one of 300 bp flanks

Page 20. “After the initial analysis we also filtered out uncommon ...” – did it affect all NSI calls? Did you end up with 1828 NSIs after this filtering? Please be more specific here.

Reviewer #2 (Remarks to the Author):

The paper by Wong et al. reports on the identification and analysis of non-reference sequence variants from 17 individuals. The study was based on linked-read data and the individuals were selected from diverse populations. This design allowed the identification of 2.1Mb of non-reference sequences together with their precise anchoring positions in the reference genome. To my knowledge, no other study has identified this much of human non-hg38 sequence with precise anchoring positions before. The calling approach is both assembly- and alignment-based, which is computationally expensive but conducive. Further, the authors confirmed previous findings using their call set, indicating that their call set is of high quality. One of these findings is that the majority of the polymorphic non-reference sequences are ancestral as they are present in other primate species. In my opinion, the most interesting new results are insights on the content of transposable elements within and around non-reference sequence variants, which was possible to examine thanks to the design of the study using linked-read data. Further, population stratification based on non-reference sequence is convincingly demonstrated and indications of transcription are given. The main conclusion picks up a suggestion from Kehr et al. (2017) that the non-reference sequences that are common across diverse populations, which are shown to exist in this paper, should be included in the reference genome. Overall, I think that the study was well-designed and the paper describes a valuable resource with interesting results for a wide community. Still, I have a few concerns about specific parts of the methodology of the call set analysis and obvious mistakes in specific conclusions, which need to be fixed.

Specific comments:

(1) Figure 3d possibly conveys a wrong conclusion, namely the presence of two ARMD hotspots within the Alu consensus sequence. Previous related studies identified only a single hotspot. If I understand the methods section "Determining transposable element[s] flanking NSIs" correctly, the analysis overcounts events where the breakpoints are ambiguous over a large range of positions. Consequently, regions that are more conserved between different Alu elements may appear as hotspots without actually being hotspots. Each variant should only contribute a total of 1 to the histogram in Figure 3d, i.e., only $1/\text{length_of_range}$ should be added per position if an event is counted for all positions in the range. Does the second "hotspot" disappear if the analysis is done correctly?

Why were only those events included where there is a 20-100 bp overlap in the alignments? The given reason ("Alu elements that had significantly diverged from one another") does not make sense to me. A single error may break off the alignment but does not indicate divergence of the Alu elements. Also, if significant divergence was needed, events with an overlap of <20bp should be included.

(2) There is a mistake in the discussion of the comparison to the Icelandic data. It states "that only 17 individuals [...] were needed to capture more than a third of the novel sequences initially identified in over 15,000 Icelanders." But only 574 (~14,5%) out of 3971 Icelandic variants were identified here. Correct is that more than a third of the variants found here were also found in the 15,000 Icelanders, which is a different statement.

Furthermore, the founder effect mostly leads to elevated frequencies of some variants that are rare in other populations. I would not expect many non-reference sequence variants to be unique to Icelanders as the majority seems to be ancestral, thus, likely shared with other populations.

(3) The results and discussion are mostly separated in the manuscript. In some places, the findings would be easier to understand if the results were put into context immediately. For example on page 11: The result that AluX and AluY are the most abundant players in recombination events is expected and uninteresting for readers who know that these are the most abundant Alu elements in the human genome. Those readers who don't know might be misled here. Another example is on page 7 and in Figure 2a, where the number of alignments per species reflects the primate phylogeny.

(4) As I am not an expert on gene expression and ESTs, I don't feel qualified to finally judge about the validity of all claims on NSI expression. I would welcome more discussion and context here, e.g. is 7% a lot or expected? And why? Furthermore, in the methods section "Aligning RNA sequencing reads to the NSI call set" I was irritated why BWA-MEM was used to align RNA-Seq reads to the "reference set". Is the "reference set" the reference genome? Why was BWA-MEM and not an RNA-Seq alignment program used that may be able to align reads across exon-exon junctions? What is the definition of unmapped reads in this analysis, did you allow large insert sizes here that are due to exon-exon junctions? Did you exclude the 300bp flanking regions when counting the number of aligned reads per NSI? Would a better way to do this analysis be to insert the non-reference sequences into the reference genome, align the RNA-Seq reads to this modified reference, e.g. with the program STAR, and then examine the level of NSI expression?

(5) Some details in the methods are missing or wrong:

- What do you mean by "correct insert size"? I assume a read pair was considered unmapped if it was *not* mapped within some range around the median insert size, but the text states "Read pairs mapped within insert sizes". Also, what range was used?

- What barcode length was used for linking the reads? Trimming off only 17bp seems very little to me in case the barcode was 16bp long. This would mean the "N-mer oligo" is only a single base pair long.

- In the section "Identification of read clusters": Did you really calculate the genome-wide coverage? How was the genome-wide coverage used? Or did you maybe calculate the coverage in windows of a certain size?

- How can contigs be separated by 200 bp?

- What does the "custom script" do that was used to compute the precise breakpoints? Does it compute a split alignment or does it rely on the Lastz alignments?

- Why did you filter sequences with an overlap of 800bp or more on the reference genome? These may have been duplications separated by unique sequence, thus matching the definition of NSIs. Did you maybe observe artifacts in the data?

- When concatenating the call sets from the 17 individuals, how did you remove duplicates?

- Why were "the homozygous calls not definitive"? What is the reason that you do not trust the haplotype assemblies? Did you consider to re-genotype the unified call set in all 17 individuals?

- In the validation using BioNano optical maps, what set of reported SVs was used?

- There is a mistake in the section "Alu recombination-mediated size distributions": The described events are deletions from the perspective of evolution, not "from the perspective of the human reference genome".

- Why did you require at least 3 carriers of an NSI for some analyses and why are 2 carriers not enough? I agree that evidence for NSIs with a single carrier is weak, but when you observe the same event in two individuals, it is in my experience well-supported. In particular, when you compare to the Icelandic data, the NSIs only present in European individuals may be shared with the Icelanders. Icelanders are of Norwegian and British origin and one of the 17 individual is British.

- Can you provide the command lines for the programs used, i.e. LongRanger, Supernova, BWA-MEM, Lastz, BLAST, bedtools, RepeatMasker, duskmasker?

(6) The methods section needs some proofreading, especially towards the end. I spotted the following typos/have the following suggestions:

- Page 16: "a BioNano optical map" not "maps"

- Page 16: "In other words" not "In another words" ... "BioNano SV calling is error-prone where multiple SVs occur in tandem"

- Page 17: "in each NSI" not "in each NSIs"

- Page 18: "Alu element consensus" not "Alu elementconsensus"

- Page 18: "NSIs flanked by Alus on both ends" instead of "NSIs with Alus flanking on both ends"

- Page 18: "to which the NSIs" not "NSI"

- Page 19: "to hg38" not "to the hg38"
- Page 20: "optical maps" not "optimal maps"
- Page 20: "All custom scripts used in this paper" instead of "All codes used in this paper"

Further comments:

- Page 4: The abstract is essentially repeated as the last paragraph of the introduction, almost word by word.
- Page 3: The statement in the introduction that "SVs play a greater role in contributing to genome diversity than SNPs" is very strong. They certainly play a major role and have been shown to affect more base pairs than SNPs. But since already a single SNP can disrupt the entire function of a gene, I think that it is difficult to measure whether SNPs or SVs play a greater role.
- Page 4: "low to medium coverage" is vague, why not say "below xx-fold coverage".
- Page 4: Long range sequence information is not necessary for reconstructing NSIs without repeats as demonstrated by previous call sets.
- Page 10: The human reference genome is dominated to 70% by a single individual of African-European admixed ancestry (Schneider et al., 2017).
- Page 12: For me "population-wide" means within one population. I guess you mean a survey across diverse populations?
- Page 14: BWA assigns a mapping quality of 0 to reads with two or more equally good mapping locations, as far as I know. This would mean that a read with mapping quality below 30 should not necessarily be included in the set of unaligned reads.
- Figure 2b: Why not add the percentage of NSIs found in chimp per number of populations to highlight that NSIs shared by more populations are more often found in chimp? In the figure caption it is unclear what the Chi² test is for. This only becomes clear when reading the main text.
- Figure 3c: The drawing suggests that the blue sequence occurs in the green and red Alu sequence but at different positions. Is this correct?
- I know that "novel sequence insertion" is commonly used to describe variants of previously unknown sequence but think that it can be misleading (the sequences were most likely not newly generated and inserted into the genomes but rather they are ancestral and have been deleted). "Non-reference sequence variants" would be more accurate in my opinion.
- I cannot find the sequencing read data under the provided accession numbers. Will the data be made freely accessible upon publication?

Reviewer #3 (Remarks to the Author):

This paper describes an analysis of 17 human genomes, sequenced with Linked-Read technology, to find novel insertion sequences.

I only evaluate the methodology, and the methods here are rather straightforward. Since this paper is not about methods development, but to build the insertion sequences, I will not object to the general framework.

- The paper is written in passive voice. Consider to edit it and change everything to active. In this form it is not obvious what the authors did themselves, rather than obtaining from other sources. For example they downloaded 3 genomes from 10x Genomics, and they say "sequencing reads were aligned". Did the authors align, or did they download pre-aligned BAM files?

- Nature Communications is open access journal, so the data and tools should be open access. Please release your "custom-built pipeline" (even though it is straightforward), and the data set from the 14 genomes you sequenced.

- Authors use optical mapping data to validate the insertions. Where is this data? Did the authors generate it? If so, release.

- There is no clear indication that the novel sequences are screened for non-human genome contamination, which is a known problem in de novo sequencing (bacteria, virus, plants, yeast, etc.).

- page 5: "All NSI sequences from 17 samples were concatenated". Did you mean "merged", or "collated"? Concatenation means something else.

- There are no references to Table 1. It is there, it should be there, but nowhere in the text it is mentioned.

Methods section:

- NSI discovery starts with filtering some reads. In page 13, it says: "Read pairs mapped within insert sizes". It should be --outside-- insert sizes.

- "Reads with an alignment score ≤ -80 ". What does "score" mean here? Do you mean the length of the alignment? This cannot be MAPQ. Also, why is it a negative number?

- page 14: again score and 90 and 113. Why 90 and 113, and why 80 above? Has to be related to read length, which is not even mentioned here. Give a read-length fraction instead of magic numbers.

Minor:

- please change all occurrences of "linked reads" to "Linked-Reads", as this is the format 10xG uses.

- repeatMasker -> RepeatMasker

- bam -> BAM

- sam -> SAM

- fastq -> FASTQ

- page 5: "another 3 samples" "an" vs "3"

May 23, 2018

We would like to thank the three reviewers for their detailed reviews and constructive feedback. Per their suggestion, we have changed the term “Novel Sequence Insertion” (NSI) to “Non-reference Unique Insertion” (NUI) to accurately reflect the nature of these sequences.

To make sure that the software code deposited to the public database indeed produce the data reported in the manuscript, we reran the analysis for all the samples and we discovered that the analysis of one sample (HG00353) was aborted midway through the original pipeline run. The number of NUI for this sample reported initially was incorrect. We reported 554 NUIs in HG00353 in the previous manuscript, but upon rerunning the pipeline on HG00353 end-to-end, the number increased to 663. Minor inconsistencies due to downstream filtering were also spotted in several other samples. However, none of the conclusions have changed due to these negligible inconsistencies and we have corrected all the reported values to reflect the changes.

Detailed responses to the reviewers' comments are as follows:

Reviewer #1:

Manuscript is poorly written; it feels like reading lab journal. Large parts of it explains obvious concepts (e.g. details on how optical mapping works on page 16 or long discussion on minor alleles in genome and divergence among primate species on page 10). The size of the manuscript should be significantly shortened.

We appreciate the reviewer's assessment and have made changes to shorten the manuscript and focus on the biological implications of our work. In particular, the long discussion on minor alleles and primate phylogeny has been truncated to make the text more concise.

At the same time, some other parts of manuscript are incomplete or formulated ambiguously. I would, therefore, suggest that a major revision is needed to bring this interesting research to good shape.

We have taken the reviewer's point to heart and have made the changes according to the very helpful suggestions.

Page 3. Authors use term Novel Sequence Insertion, while they also show their presence in genomes of other primates. I suggest to use term Non-reference Sequence Insertion instead.

We agree that the designation NSI does not reflect the true nature of the sequences in question but the term Non-reference Sequence Insertion is somewhat misleading because it usually implies highly repetitive sequences that are derived from the telomeres, centromeres, or the short-arms of the acrocentric chromosomes. We have changed the term to Non-reference Unique Insertion (NUI) instead to emphasize that they are unique sequences rather than repetitive sequences not assembled into the reference genome.

Page 5. References to data URLs (10G website), versions of software (Long ranger and Supernova) should be given, preferably in methods section.

We thank the reviewer for the reminder. The relevant information has been added to the first paragraph of the methods section.

Page 5. Pearson correlations 0.28 and 0.26 do NOT indicate that “NSI were not biased”, rather they do not show evidence for such bias.

The wording has been changed according to the reviewer's correction.

Page 6. "average recall rate 88.2%" – you need to mention average number of validated events per sample in the main text.

The number has been added to the second paragraph under "NUI discovery pipeline and validation" in the results.

Page 6. "Americans are admixed with South Asians" – do you mean 'cluster with' here?
Yes. Accordingly, the wording has been changed in the text.

Page 7. "being conserved among all four non-human primates" – do you mean 'present in genomes of all four...'

The language has been changed in the text per the reviewer's suggestion.

Page 7. "NSI aligned to the chimpanzee genome had any underlying population structure" – too vague

A more precise description of the analysis has replaced this sentence. The sentence now reads as follows: "Next, we assessed whether the more common NUIs exhibit enrichment for ancestral sequences."

Page 7. "NSI ... were significantly enriched in chimpanzee" – same as above, be more specific.

A sentence has been added to underline the magnitude of this enrichment.

Page 7. "NSI sequence characterization" – too broad header for a RepeatMasker analysis. Something like 'analysis of repeat sequences in NSIs' would better reflect the essence of that section.

We thank the reviewer for the suggestion. The header for the RepeatMasker analysis has been changed.

Page 8. Give a reference upon first appearance of ARMD.

Done.

Page 8. Further you claim that NSIs are caused by ARMD, while this is a likely explanation for their origin and you do not have proof the mechanism was ARMD, e.g. "... Alu subspecies responsible for ARMD in this analysis".

We appreciate the reviewer's point. The language of the sentence has been changed to "Alu recombination-mediated deletion (ARMD) is a likely driver of this observation."

Page 8. Further, one of the ways to support ectopic recombination mechanism would be to check ancestral states of these loci in chimp, gorilla and bonobo.

Our data showed that out of 265 total NUIs that appeared to have emerged through ARMD, 163, 167, 102, and 38 events were found in chimpanzee, gorilla, bonobo, and orangutan respectively (Supplementary Fig. S3). This new information is now incorporated under "Analysis of repeat sequences in NUIs" in the results section.

Page 8. "Biological consequences of NSI" – again too broad title when you simply check for expression evidence.

The title has been changed to "Transcription potential of NUIs".

Page 9. Might be a good idea to compare Icelandic NSIs to those that you identified in European population.

When we compared the Icelandic NUIs to those we identified in the five human populations, we found that almost 90% of the shared NUIs were ancestral sequences. Hence, we did not expect the Europeans in our study cohort to share a significantly large number of sequences with the Icelandic data set. We created a new boxplot to depict the population structure of these shared NUIs (**Supplementary Figure 4**) and the plot largely resembled Figure 1c in the main manuscript. Specifically, Africans had the highest number of shared NUIs while Europeans had the fewest. This new piece of data has been added to the supplementary information.

Page 10. “patterns of SVs” – patterns of genomic inserts
Fixed.

Page 10. “dominated by SEQUENCES FROM DNA donors”
Fixed.

Page 10. “NSI could be used to distinguish the populations” – too impractical

The original intent of this statement was to show that the small sample size didn't interfere with the population analysis because PCA could faithfully recapitulate population pattern and admixture. The sentence has been changed to avoid such confusion. The change is highlighted in the first paragraph of the discussion section.

Page 10. “people of different origins” – ‘people of different ethnic backgrounds’
Fixed.

Page 10. “Having minor alleles ... can interfere with every stage” – it is my understanding that insertions have preference over deletions for being included in genome reference. That leads to more complete genome reference.

We thank the reviewer for these comments. A similar description is now added to the last sentence of the first paragraph in the discussion section.

Page 11. About half of our genomes are repeats. Did you check for significant overrepresentation of Alus/LINEs in NSIs?

We thank the reviewer for this suggestion. Analysis with RepeatMasker showed that 21.4% and 23.4% of the NUIs were SINEs and LINEs respectively (**Supplementary Table S3**). Previous analysis of the repeat content reveals that 13% and 21% of the entire human genome is composed of these two respective elements (Lander et al., 2001). These numbers indicate that the SINEs are overrepresented in the NUIs but not the LINEs. Based on the work presented here, insertions were required to have at least 50 non-masked bases to be included as NUIs. LINE- or *Alu*-mediated insertional mutagenesis may contribute to the current dataset but this mechanism does not necessarily introduce novel sequences to the genome. Since we observed a disproportionately high frequency of NUIs that were results of *Alu*-recombination mediated deletions, we speculate that this subset of the alignments is a likely cause of the enrichment. A similar description is also added in the second paragraph of the discussion section.

Page 11. “ectopic recombination ... MIGHT BE responsible for about 13% of all NSIs”
The language has been changed.

Page 11. “In addition the putative recombination hotspots ...” – the discussion of this fact is bigger and properly referenced, while I would expect that in results section.

Per the reviewer's suggestion, the text has been rearranged so that this is now in the results section.

Page 13. “read pairs mapped within insert sizes” – this does not make sense to me.

Thanks for pointing out this mistake. Read pairs that were NOT mapped within the insert sizes were extracted for downstream analysis. BWA has a built-in algorithm to estimate the correct insert size based on the distribution of the read pairs. For mapping short reads, an incorrect insert size is usually about 6-7 standard errors away from the mean mapping distance between a read pair. More details can be found in the BWA manual.

Page 14. “Contigs whose ends were separated by less than 200bp are merged” – not sure what that means

After extending 7kb to each end of the sequences as anchors, if the ends between two different contigs were overlapping or separated by less than 200bp, the entire sequence from the upstream left anchor to the downstream right anchor was extracted for Lastz alignment. This step was implemented to simplify the downstream filtering procedures for large SVs. The sentence has been clarified under “Identification of read clusters” in the methods section.

Page 15. “Finally, we concatenated all the NSIs” – do you mean ‘overlapped’?

We thank the reviewer for the correction. The word is now changed to ‘overlapped’.

Page 17. Definition of recall rate is incorrect, formula you are giving calculates precision. What did you plan to calculate?

We agree that we misused the term “recall rate” in the original manuscript. We calculated the number of validated events based on BioNano data divided by the total NUI count. We have replaced the equation to the following:

$$\text{precision} = \frac{\text{NUIs supported by BioNano}}{\text{NUIs supported by BioNano} + \text{NUIs not supported by BioNano}}$$

Page 19. When checking RNA-seq reads in NSIs – do you check that read pair does not fully reside in one of 300 bp flanks

Yes, we ensured that both reads from the read pairs cannot fully reside in the 300bp flanks.

Page 20. “After the initial analysis we also filtered out uncommon ...” – did it affect all NSI calls? Did you end up with 1828 NSIs after this filtering? Please be more specific here.

As requested from Reviewer 3, we have changed the methods so that we are now defining uncommon NUIs as those that are found in only one individual. In other words, all singletons are now removed from this part of the analysis. After this filtering step, 1,308 NUIs remained. All calculations that followed used 1,308 as the denominator.

Reviewer #2:

Figure 3d possibly conveys a wrong conclusion, namely the presence of two ARMD hotspots within the Alu consensus sequence. Previous related studies identified only a single hotspot. If I understand the methods section “Determining transposable element[s] flanking NSIs” correctly, the analysis overcounts events where the breakpoints are ambiguous over a large range of positions.

Consequently, regions that are more conserved between different Alu elements may appear as hotspots without actually being hotspots. Each variant should only contribute a total of 1 to the histogram in Figure 3d, i.e., only 1/length_of_range should be added per position if an event is counted for all positions in the range. Does the second “hotspot” disappear if the analysis is done correctly?

We appreciate the reviewer’s suggestions and we agree that the proposed method can more accurately define ARMD hotspots. We changed the methodologies as described. Briefly, by adding only 1/length_of_range per position across all events with 10-290bp overlap in the alignments, the

plot largely resembled the previous plot in which two hotspots were observed. The first hotspot was very pronounced while the second one was not as striking. Although the previous study (Han et al., 2007) only identified a single hotspot, the dataset they analyzed did show a mild elevation in the percentage of breakpoint at position 156-205bp of an *Alu* element. Figure from the publication is attached.

Figure: Han et al., 2007

Why were only those events included where there is a 20-100 bp overlap in the alignments? The given reason ("Alu elements that had significantly diverged from one another") does not make sense to me. A single error may break off the alignment but does not indicate divergence of the Alu elements. Also, if significant divergence was needed, events with an overlap of <20bp should be included.

Initially we reasoned that *Alu* elements with significant overlaps might not require these hotspots for ARMD, hence they were excluded from the original analysis. In addition, adding the breakpoints at position 1 and position 290 would greatly inflated the frequency line. However, using the approach suggested above mitigated this bias (the two ends no longer have very high breakpoint frequency). Thus, we have added those alignments to the analysis described above (and in the revised methods). We initially picked 20bp as the lower bound since the published work (Han et al., 2007) showed a 22bp hotspot that seems to mediate ARMD. We later reasoned that this requirement might not be necessary and shorter hotspots may exist. We thus further lowered this minimum size to 10bp. All ARMD related analysis now use this 10bp threshold. Alignments with an *Alu* overlap shorter than 10bp created sharp jagged peak in the plot but the overall plot shape did not change.

There is a mistake in the discussion of the comparison to the Icelandic data. It states "that only 17 individuals [...] were needed to capture more than a third of the novel sequences initially identified in over 15,000 Icelanders." But only 574 (~14,5%) out of 3971 Icelandic variants were identified here. Correct is that more than a third of the variants found here were also found in the 15,000 Icelanders, which is a different statement.

We thank the reviewer for the correction. The statement has been changed to accurately reflect the dataset.

Furthermore, the founder effect mostly leads to elevated frequencies of some variants that are rare in other populations. I would not expect many non-reference sequence variants to be unique to Icelanders as the majority seems to be ancestral, thus, likely shared with other populations.

The reviewer's input on this subject is greatly appreciated. We have changed the language to emphasize that only a small proportion of the observed genomic alterations would be explained by the founder effect. The change is highlighted in the fourth paragraph of the discussion section.

The results and discussion are mostly separated in the manuscript. In some places, the findings would be easier to understand if the results were put into context immediately. For example on page

11: *The result that AluX and AluY are the most abundant players in recombination events is expected and uninteresting for readers who know that these are the most abundant Alu elements in the human genome. Those readers who don't know might be misled here. Another example is on page 7 and in Figure 2a, where the number of alignments per species reflects the primate phylogeny.* Per the reviewer's helpful suggestions, we have rearranged the text to make the claims more cohesive throughout the paper.

As I am not an expert on gene expression and ESTs, I don't feel qualified to finally judge about the validity of all claims on NSI expression. I would welcome more discussion and context here, e.g. is 7% a lot or expected? And why?

The human dbEST can be used for various purposes but one of its main use is to identify novel genes. EST sequences are typically 200-500bp long and with only one end of an mRNA product sequenced. Entries in other mRNA databases are annotated based on the human reference genome and they are not useful in the identification of novel exons. While it is difficult to draw a comparison in this context, we believe that 7% is significant given the fact that the database itself is not comprehensive and only the ends of mRNAs are typically sequenced. Additionally, when we compared this result with that of RNA-seq reads (the orthogonal approach), the overlap of transcribed NUIs was statistically significant (Fisher's $p = 8.018409e-17$; OR = 4.83). This means that if an NUI was found to be expressed in dbEST, it's almost 5 times more likely to be identified as expressed in the RNA-seq analysis relative to the null.

Furthermore, in the methods section "Aligning RNA sequencing reads to the NSI call set" I was irritated why BWA-MEM was used to align RNA-Seq reads to the "reference set". Is the "reference set" the reference genome? Why was BWA-MEM and not an RNA-Seq alignment program used that may be able to align reads across exon-exon junctions?

BWA-MEM was used because our initial analysis did not require knowing the precise positions of the splicing junctions. The RNA-Seq aligner TopHat also uses Bowtie (a non-RNA-seq aligner) to perform short read alignment due to its speed and accuracy. TopHat then uses the output to refine the splicing junctions. For our purposes, we only wanted to know whether an RNA-seq read was mapped to the genome and the accurate splicing information was unnecessary. In the scenario in which an RNA-Seq read spans over exon boundary, the read will be mapped contiguously until the exon junction, and at that point the rest of the read will be clipped off. This read will have a poor mapping quality and a poor alignment score, but the read will not be considered unmapped.

While we strongly believe that BWA can give us reliable alignment results, we reason that we should also try to rerun the same analysis with an RNA-Seq aligner to validate our findings. Using the RNA-seq aligner STAR, we obtained nearly double the alignment hits. This result was not surprising owing to the fact that the STAR aligner is more conservative during the first alignment step. We saw a threefold increase in the number of unmapped RNA-Seq reads when using STAR. While extracting unmapped reads for downstream processing, we carefully removed reads that were deemed unmapped due to having too many mapped loci. Extracted reads were then mapped to the NUIs with 300bp flanking on both ends using STAR. Further details can be found in the methods section of the manuscript.

Results using the STAR aligner has replaced those found in the original text in both the results and the methods sections. We note that the combination of transcribed NUIs from the dbEST and the RNA-seq methods (STAR) is no longer significantly enriched in ancestral sequences (Fig.4b of the original manuscript) and the claim is now removed from the revised manuscript.

What is the definition of unmapped reads in this analysis, did you allow large insert sizes here that are due to exon-exon junctions?

The SAM flag 4 was used to extract unmapped reads. Whenever exon-exon junctions separate read pairs with large insert sizes, both reads from the pairs would still be mapped but they won't be mapped as proper pairs. However, since we only extracted reads that didn't map at all, size of exon-exon junctions would not interfere with the results.

Did you exclude the 300bp flanking regions when counting the number of aligned reads per NSI? Would a better way to do this analysis be to insert the non-reference sequences into the reference genome, align the RNA-Seq reads to this modified reference, e.g. with the program STAR, and then examine the level of NSI expression?

We made sure that both reads from the pairs did not fully reside in the 300bp flanking sequences. While this is not the only approach to perform this analysis, we believe that inserting the NUIs into the reference genome wouldn't have added benefits, especially since the reads have already been mapped to the reference before.

Some details in the methods are missing or wrong:

*- What do you mean by "correct insert size"? I assume a read pair was considered unmapped if it was *not* mapped within some range around the median insert size, but the text states "Read pairs mapped within insert sizes". Also, what range was used?*

Thanks for pointing out this mistake and the description in the main text has been corrected. Read pairs that were NOT mapped within insert sizes were extracted for downstream analysis. BWA has a built-in algorithm to estimate the correct insert size based on the distribution of the read pairs. For mapping short reads, an incorrect insert size is usually about 6-7 standard errors away from the mean mapping distance between a read pair. More details can be found in the BWA manual.

- What barcode length was used for linking the reads? Trimming off only 17bp seems very little to me in case the barcode was 16bp long. This would mean the "N-mer oligo" is only a single base pair long.

That was a mistake in the original manuscript and is now fixed. The pipeline actually trimmed off a total of 23bp from read1—16bp of barcode and 7bp of low accuracy sequence from the N-mer oligo.

- In the section "Identification of read clusters": Did you really calculate the genome-wide coverage? How was the genome-wide coverage used? Or did you maybe calculate the coverage in windows of a certain size?

The word genome was used loosely to denote haplotype. The "haplotype-wide" coverage is calculated with bedtools using the following command:

```
> bedtools genomecov -ibam /path/to/bwa/output/filtered/SAM -bg > out.bed
```

As stated in the methods section, the sequences corresponding to the positions with coverages between 8 to 100 were extracted (sequence extraction also included the 7kb anchor on each end).

- How can contigs be separated by 200 bp?

We apologize for the confusing wording here. After extending 7kb to each end of the sequences as anchors, if the ends between two different contigs were overlapping or separated by less than 200bp, the entire sequence from the upstream left anchor to the downstream right anchor was extracted for Lastz alignment. This step was implemented to simplify the downstream filtering procedures for large SVs.

- What does the "custom script" do that was used to compute the precise breakpoints? Does it compute a split alignment or does it rely on the Lastz alignments?

Breakpoint was determined based on the lastz alignment output. Breakpoint positions were computed by identifying positions where the sequence alignment broke off and realigned.

- Why did you filter sequences with an overlap of 800bp or more on the reference genome? These

may have been duplications separated by unique sequence, thus matching the definition of NSIs. Did you maybe observe artifacts in the data?

We noticed in the past that alignments that created an overlap of 800bp had poor concordance rate with the BioNano insertion calls. Even when BioNano makes an insertion call at the corresponding genomic locus that overlaps an NUI candidate, the median size difference between BioNano prediction and the length of that NUI was about 3kb. In contrast, the median size difference for all other alignments were usually between 300-600bp. While BioNano cannot determine the precise insertion size due to its inherent resolution limits, this difference suggested that the sequences with large overlap were not as reliable. To ensure our data were of high quality, we decided to discard all alignments with such a large overlap.

- When concatenating the call sets from the 17 individuals, how did you remove duplicates?

Duplicates were defined as either having one identical breakpoint shared with another individual or two breakpoints (start and end) that were both within 50bp from a different individual.

- Why were "the homozygous calls not definitive"? What is the reason that you do not trust the haplotype assemblies? Did you consider to re-genotype the unified call set in all 17 individuals?

We believe that some of the homozygous calls were made incorrectly due to our observations that many of the NUI singletons are homozygous calls rather than heterozygous calls. We believe that, in these cases, the supernova assembler occasionally places the NUIs in both pseudo-haplotypes incorrectly. When this occurs, we could not find the reference sequence in any of the two pseudo-haplotypes but the alignment file might have perfectly mapped reads that spanned the corresponding breakpoints without clippings or gaps. This indicates that the reference allele at the locus of interest is present but supernova is unaware of this other allele.

We currently do not have any plans to re-genotype the call set because we believe that is not within the scope of this paper. The focus of the current study is to get a general sense of the magnitude of NUI diversity across different human populations. We also set out to identify the origin and mechanism of novel sequences in the human genome. We strongly believe that re-genotyping would be important when we start building a new reference genome by incorporating sequences from diverse populations, but for now, knowing the exact genotype (whether the NUIs are het or alt homo) are not necessary to achieve these immediate goals.

- In the validation using BioNano optical maps, what set of reported SVs was used?

Only insertions were used to validate the NUIs. The clarification is highlighted in the methods section under "NUI validation using BioNano optical maps".

- There is a mistake in the section "Alu recombination-mediated size distributions": The described events are deletions from the perspective of evolution, not "from the perspective of the human reference genome".

We thank the reviewer for the correction. The change has been made and highlighted in the methods section under "ARMD size distributions".

- Why did you require at least 3 carriers of an NSI for some analyses and why are 2 carriers not enough? I agree that evidence for NSIs with a single carrier is weak, but when you observe the same event in two individuals, it is in my experience well-supported. In particular, when you compare to the Icelandic data, the NSIs only present in European individuals may be shared with the Icelanders. Icelanders are of Norwegian and British origin and one of the 17 individual is British.

Three carriers were required in the initial analysis because we wanted to have a set of well-supported NUIs for comparisons. However, we agree that this is not necessarily the best way to analyze our

dataset. We have recomputed the percentages per the reviewer's recommendation by only filtering out NUIs with just one occurrence. Please see the revised text for the updated results.

- Can you provide the command lines for the programs used, i.e. LongRanger, Supernova, BWA-MEM, Lastz, BLAST, bedtools, RepeatMasker, dustmasker?

Yes. The pipeline source code is also deposited on GitHub (https://github.com/wongkarenhy/NUI_pipeline.git).

Longranger:

```
>longranger wgs --fastqs=/path/to/fastqs --id=sample_longranger --reference=/path/to/reference/refdata-GRCh38-2.1.0 --jobmode=sge
```

To start the supernova de novo assembly:

```
>supernova run --id=sample_sn --fastqs=/path/to/fastqs
```

To generate the assembly output:

```
>supernova mkoutput --asmdir=/path/to/assembly/sample_sn/outs/assembly --outprefix=sample --style=pseudohap2
```

BWA-MEM:

```
>bwa mem -t 32 -p /path/to/pseudohap.fa /path/to/sorted/fastq
```

Lastz:

```
>lastz_32 /path/to/reference/hg38_core_chrs.fa[multiple][unmask] /path/to/contig/fasta[nameparse=darkspace] --format=general:number,name1,start1,end1,length1,size1,name2,start2+,end2+,length2,size2,strand2,score,nmatch,nmismatch,cigar,identity,cov% --step=20 --seed=match15 --notransition --exact=400 --identity=95 --match=1,5 --ambiguous=iupac > out
```

BLAST:

```
blastn -query /path/to/fasta \
      -db /path/to/reference/fasta \
      -task megablast \
      -dust no \
      -outfmt "7 qseqid sseqid evalue bitscore qlen pident length salltitles qstart qend sstart send nident mismatch gapopen gaps qcovs qcovhsp" \
      -max_target_seqs 1 \
      -max_hsps 1 \
      -out /path/to/output.blast \
      -num_threads 32
```

Bedtools:

```
> bedtools genomecov -ibam /path/to/bwa/output/filtered/SAM -bg
```

RepeatMasker:

```
> RepeatMasker --species human -pa 32 in.fa
```

dustmasker:

```
> dustmasker -outfmt fasta -in in.fa -out out.fa
```

Page 16: "a BioNano optical map" not "maps"

Noted and corrected.

- Page 16: "In other words" not "In another words" ... "BioNano SV calling is error-prone where multiple SVs occur in tandem"

Noted and corrected.

- Page 17: "in each NSI" not "in each NSIs"

Noted and corrected.

- Page 18: "Alu element consensus" not "Alu elementconsensus"

Noted and corrected.

- Page 18: "NSIs flanked by Alus on both ends" instead of "NSIs with Alus flanking on both ends"

Noted and corrected.

- Page 18: "to which the NSIs" not "NSI"

Noted and corrected.

- Page 19: "to hg38" not "to the hg38"

Noted and corrected.

- Page 20: "optical maps" not "optimal maps"

Noted and corrected.

- Page 20: "All custom scripts used in this paper" instead of "All codes used in this paper"

Noted and corrected.

Reviewer #3:

- Page 4: *The abstract is essentially repeated as the last paragraph of the introduction, almost word by word.*

We have revised the abstract to make the text more concise.

- Page 3: *The statement in the introduction that "SVs play a greater role in contributing to genome diversity than SNPs" is very strong. They certainly play a major role and have been shown to affect more base pairs than SNPs. But since already a single SNP can disrupt the entire function of a gene, I think that it is difficult to measure whether SNPs or SVs play a greater role.*

We appreciate the reviewer's point. The wording has been changed to tone down the language.

- Page 4: *"low to medium coverage" is vague, why not say "below xx-fold coverage".*

The vague language has been removed in the sentence.

- Page 4: *Long range sequence information is not necessary for reconstructing NSIs without repeats as demonstrated by previous call sets.*

Per the reviewer's comment, we have changed the sentence to stress the fact that long range sequence information is necessary to reconstruct NSIs with repetitive genomic content. This is particularly important since half of the human genome is composed of repeats.

- Page 10: *The human reference genome is dominated to 70% by a single individual of African-European admixed ancestry (Schneider et al., 2017).*

The reference is now inserted into the main text.

- Page 12: *For me "population-wide" means within one population. I guess you mean a survey across diverse populations?*

Yes, we have replaced the term "population-wide" in the main text.

- Page 14: *BWA assigns a mapping quality of 0 to reads with two or more equally good mapping locations, as far as I know. This would mean that a read with mapping quality below 30 should not necessarily be included in the set of unaligned reads.*

This is correct. Reads that can align perfectly well to multiple locations shouldn't be included. We did not keep reads if the mapping quality was below 30.

- Figure 2b: *Why not add the percentage of NSIs found in chimp per number of populations to highlight that NSIs shared by more populations are more often found in chimp? In the figure caption it is unclear what the Chi² test is for. This only becomes clear when reading the main text.*

Per the reviewer's suggestion, we have added the percentages to figure 2b.

- Figure 3c: *The drawing suggests that the blue sequence occurs in the green and red Alu sequence but at different positions. Is this correct?*

Yes, this is correct.

- *I know that "novel sequence insertion" is commonly used to describe variants of previously unknown sequence but think that it can be misleading (the sequences were most likely not newly generated and inserted into the genomes but rather they are ancestral and have been deleted). "Non-reference sequence variants" would be more accurate in my opinion.*

We appreciate the reviewer's suggestion but the suggested term "Non-reference Sequence Variant" commonly refers to highly repetitive sequences that are derived from the telomeres, centromeres, or

the short-arms of the acrocentric chromosomes. We have changed the term to Non-reference Unique Insertion (NUI) to reflect the unique nature of the sequences. We intentionally kept the word “insertion” because we would like to emphasize that these are insertion relative to the human genome reference as the terms, “insertion” and “deletion” have always been used traditionally to describe events that differ from the human reference genome.

- I cannot find the sequencing read data under the provided accession numbers. Will the data be made freely accessible upon publication?

Yes, all data will be made publicly available upon acceptance.

- The paper is written in passive voice. Consider to edit it and change everything to active. In this form it is not obvious what the authors did themselves, rather than obtaining from other sources. For example they downloaded 3 genomes from 10x Genomics, and they say "sequencing reads were aligned". Did the authors align, or did they download pre-aligned BAM files?

We thank the reviewer for pointing out our use of ambiguous languages. We have made changes to describe what we have actually done in this study.

- Nature Communications is open access journal, so the data and tools should be open access. Please release your "custom-built pipeline" (even though it is straightforward), and the data set from the 14 genomes you sequenced.

The data set will be made available upon acceptance. All custom scripts used in the pipeline has already been deposited on GitHub (https://github.com/wongkarenhy/NUI_pipeline.git).

- Authors use optical mapping data to validate the insertions. Where is this data? Did the authors generate it? If so, release.

BioNano assemblies for the 17 samples are included as part of BioProject PRJNA418343 and the data will be public as soon as GenBank releases the data. We have already given GenBank permission to release the data as of 5/1/2018.

- There is no clear indication that the novel sequences are screened for non-human genome contamination, which is a known problem in de novo sequencing (bacteria, virus, plants, yeast, etc.). Sequencing contaminants from bacteria, virus, plants, yeast etc will not anchor to the primary hg38 reference assembly. Additionally, contaminants will not have significant barcode sharing with nearby sequences and they will not be assembled into long contigs. However, to ensure that our sequences did not have foreign DNA, we did run a contamination screen using BLAST to ensure that these NUIs did not contain any sequences from bacteria, human microbiome, univec, and virus. The result shows that none of the NUIs had significant alignments with these common contaminants (% identity 90 and % query coverage 90). A new section titled “Contamination screen” is added in the methods section.

- page 5: "All NSI sequences from 17 samples were concatenated". Did you mean "merged", or "collated"? Concatenation means something else.

The word has been replaced to avoid confusion.

- There are no references to Table 1. It is there, it should be there, but nowhere in the text it is mentioned.

Reference to Table 1 can be found under the section “The structure of genetic diversity across populations” at the end of the first sentence.

Methods section:

- NSI discovery starts with filtering some reads. In page 13, it says: "Read pairs mapped within insert sizes". It should be --outside-- insert sizes.

We thank the reviewer. It is now fixed in the text.

- "Reads with an alignment score ≤ -80 ". What does "score" mean here? Do you mean the length of the alignment? This cannot be MAPQ. Also, why is it a negative number?

The alignment score is a SAM tag generated by Long Ranger using the Lariat alignment algorithm. The scoring parameters are $AS:f = -2 * \text{mismatches} - 3 * \text{indels} - 5 * \text{clipped} - 0.5 * \text{clipped_bases} - 4 * \text{improper_pair}$. Using Lariat, the highest possible score for an end-to-end alignment is 0.

- page 14: again score and 90 and 113. Why 90 and 113, and why 80 above? Has to be related to read length, which is not even mentioned here. Give a read-length fraction instead of magic numbers.

The scores here (90 and 113) were generated by BWA-MEM and it uses a different scoring scheme from the one shown above. Each mismatch gets a penalty of -4, and hence 90 and 113 roughly corresponded to 9 mismatches in the sequence alignment (or a combination of penalties). Similarly, an alignment score of -80 corresponds to 40 mismatches (Lariat's mismatch penalty is -2) based on the Lariat scoring scheme. This also means that if more than ~25-30% of the sequences don't align well, they will be extracted for downstream analysis.

Minor:

- please change all occurrences of "linked reads" to "Linked-Reads", as this is the format 10xG uses.

- repeatMasker -> RepeatMasker

- bam -> BAM

- sam -> SAM

- fastq -> FASTQ

- page 5: "another 3 samples" "an" vs "3"

All noted and corrected.

Reviewer #1 (Remarks to the Author):

The authors addressed my comments satisfactory. I recommend acceptance of this revised version

Reviewer #2 (Remarks to the Author):

The authors have replied to all my comments and revised the manuscript according to many of my suggestions. For example, the RNA-Seq analysis was re-done using the RNA-Seq aligner STAR, which I think has added value to the manuscript. Furthermore, the figure on ARMD hotspots has been corrected and now resembles that from previous studies more. I disagree with the authors' reply to my comment that it largely resembles the plot from the initial version of the manuscript - the second hotspot is now much less pronounced. However, I agree with their text in the manuscript that the plot reveals two hotspots.

In some cases, the authors replied to my comment without making changes to the manuscript or with insufficient changes:

- I find the comparison of the results from the dbEST and RNA-Seq analyses ($P=8e-17$ and $OR=4.83$) given in their reply to my comment very helpful for understanding the significance of these results but it was not added to the manuscript.
- The concordance rate with BioNano for NUIs flanked by duplications of different sizes should be added to the Methods section.
- The link to the pipeline's GitHub repository needs to be added to the manuscript.
- The correction of the statement in the discussion about the overlap with the Icelanders needs revision (top of page 12): The first highlighted sentence is redundant with the previous sentence and the second highlighted sentence conveys the impression that the overlap between the two data sets is 89% if singletons are removed.
- The authors misunderstood my comment about the genome-wide coverage, which again was due to my misunderstanding of their use of the word "genome-wide". For me the "genome-wide coverage" or "haplotype-wide coverage" is a single value reflecting the coverage across the whole genome. I am guessing that the authors meant to say that they calculated the coverage at all positions of the genome? Please clarify the text.
- The statement that BWA-MEM estimates the insert size based on the bulk read pair distributions doesn't necessarily mean that the BWA classification of "proper pairs" was used. It would be more

helpful to explicitly state that classification of insert sizes are taken from BWA-MEM and that BWA-MEM is called within the Long Ranger software.

- The explanation why homozygous calls are not trusted given as a reply to my comment needs to be added to the Methods section.

- Page 4: The main text still states that long-range sequence information is necessary for NUI reconstruction, and not just in repetitive regions. This could be fixed by deleting ", especially" from the sentence.

- Page 10: My comment about the reference genome being dominated to 70% by a single individual was not aiming for the addition of a reference but for improving the text, which the authors did not change but only extended. The newly added sentence sounds contradictory with the previous sentence: the first sentence is stating that the reference genome is dominated by several donors of European ancestry and the second is stating that it is dominated by a single African-European donor. To be clear, I do not disagree with the conclusion that the smaller number of NUIs found in Europeans compared to other, non-African populations may be due to the origin of the reference genome. I only think that the phrasing needs improvement.

Further comments:

- Proofreading and language editing is still needed throughout the text.

- The Icelandic data set is sometimes referred to as "the data set by deCODE genetics" and sometimes as "the Amgen data set". To avoid confusion, I suggest referring to it as the deCODE data set only.

- Page 8 "this phenomenon": What phenomenon?

- In many program commands, autocorrect has changed -- to –

Reviewer #3 (Remarks to the Author):

The authors adequately responded to my questions.

Minor:

- please include the alignment score details in the manuscript, or the Supplementary.

June 14, 2018

We are pleased that Reviewers 1 and 3 deemed our revision acceptable for publication. We are grateful to Reviewer 2 for his/her additional suggestions, including a charge to improve the readability of the paper by careful editing and proofreading. We have made significant editorial changes to the paper with help of native English speakers and we believe that this paper is much improved.

Below is the list of actions taken (in blue) to address helpful suggestions by Reviewers #2 and #3.

Reviewer #2 (Remarks to the Author):

The authors have replied to all my comments and revised the manuscript according to many of my suggestions. For example, the RNA-Seq analysis was re-done using the RNA-Seq aligner STAR, which I think has added value to the manuscript. Furthermore, the figure on ARMD hotspots has been corrected and now resembles that from previous studies more. I disagree with the authors' reply to my comment that it largely resembles the plot from the initial version of the manuscript - the second hotspot is now much less pronounced. However, I agree with their text in the manuscript that the plot reveals two hotspots.

We are pleased that the reviewer has accepted our responses to his/her main critiques.

In some cases, the authors replied to my comment without making changes to the manuscript or with insufficient changes:

- I find the comparison of the results from the dbEST and RNA-Seq analyses ($P=8e-17$ and $OR=4.83$) given in their reply to my comment very helpful for understanding the significance of these results but it was not added to the manuscript.

We have added the results to the text per the reviewer's suggestion.

- The concordance rate with BioNano for NUIs flanked by duplications of different sizes should be added to the Methods section.

Done.

- The link to the pipeline's GitHub repository needs to be added to the manuscript.

Done.

- The correction of the statement in the discussion about the overlap with the Icelanders needs revision (top of page 12): The first highlighted sentence is redundant with the previous sentence and the second highlighted sentence conveys the impression that the overlap between the two data sets is 89% if singletons are removed.

Clarified in the main text per the reviewer's suggestion.

- The authors misunderstood my comment about the genome-wide coverage, which again was due to my misunderstanding of their use of the word "genome-wide". For me the "genome-wide coverage" or "haplotype-wide coverage" is a single value reflecting the coverage across the whole genome. I am guessing that the authors meant to say that they calculated the coverage at all positions of the genome? Please clarify the text.

Clarified in the main text per the reviewer's suggestion.

- The statement that BWA-MEM estimates the insert size based on the bulk read pair distributions doesn't necessarily mean that the BWA classification of "proper pairs" was used. It would be more helpful to explicitly state that classification of insert sizes are taken from BWA-MEM and that BWA-MEM is called within the Long Ranger software.

Clarified per the reviewer's suggestion.

- The explanation why homozygous calls are not trusted given as a reply to my comment needs to be added to the Methods section.

Added to the Methods section per the reviewer's suggestion.

- Page 4: The main text still states that long-range sequence information is necessary for NUI reconstruction, and not just in repetitive regions. This could be fixed by deleting ", especially" from the sentence.

Done.

- Page 10: My comment about the reference genome being dominated to 70% by a single individual was not aiming for the addition of a reference but for improving the text, which the authors did not change but only extended. The newly added sentence sounds contradictory with the previous sentence: the first sentence is stating that the reference genome is dominated by several donors of European ancestry and the second is stating that it is dominated by a single African-European donor. To be clear, I do not disagree with the conclusion that the smaller number of NUIs found in Europeans compared to other, non-African populations may be due to the origin of the reference genome. I only think that the phrasing needs improvement.

Rephrased according to the reviewer's suggestion.

Further comments:

- Proofreading and language editing is still needed throughout the text.

We thank the reviewer for the suggestion. Done.

- The Icelandic data set is sometimes referred to as "the data set by deCODE genetics" and sometimes as "the Amgen data set". To avoid confusion, I suggest referring to it as the deCODE data set only.

Done.

- Page 8 "this phenomenon": What phenomenon?

Rephrased.

- In many program commands, autocorrect has changed -- to –

All corrected.

Reviewer #3 (Remarks to the Author):

The authors adequately responded to my questions.

Minor:

- please include the alignment score details in the manuscript, or the Supplementary.

Done.